# Quality of Fine Yarns from Modacrylic/Polyacrylate/Lyocell Blends Intended for Affordable Flame-Resistant Underwear

**DOI:** 10.3390/ma16124386

**Published:** 2023-06-14

**Authors:** Tatjana Rijavec, Mirjam Leskovšek, Neža Sukič, Barbara Rajar, Alenka Pavko Čuden

**Affiliations:** Faculty of Natural Sciences and Engineering, University of Ljubljana, Aškerčeva 12, 1000 Ljubljana, Slovenia; mirjam.leskovsek@ntf.uni-lj.si (M.L.); alenka.cuden@ntf.uni-lj.si (A.P.Č.)

**Keywords:** base layer, fire-resistant, flammability, moisture management

## Abstract

Flammability testing of undergarments is a topic that is often overlooked and rarely on the list of textiles to be tested for fire safety. However, it is particularly important for professionals exposed to fire risk to investigate the flammability of underwear as its direct contact with the skin can be critical to the extent and degree of skin burns. This research focuses on the suitability of affordable blends of 55 wt.% modacrylic, 15 wt.% polyacrylate, and 30 wt.% lyocell fibres that have the potential to be used for flame-resistant underwear. The influence of modacrylic fibre linear density (standard and microfibres), ring spinning processes (conventional, Sirospun, and compact), and knitted structure (plain, 2:1 rib, 2:1 tuck rib, single pique, and triple tuck) on their properties required for thermal comfort in high-temperature environments was investigated. Scanning electron and optical microscopy, FT-IR spectroscopy, mechanical testing, moisture regain, water sorption, wettability, absorption, DSC, TGA, and flammability were tested to assess the desired suitability. The wetting time (5–14.6 s) and water absorption time (4.6–21.4 s) of the knitted fabrics have shown excellent ability to transport and absorb water compared to the knitted fabrics created from a conventional blend of 65% modacrylic and 35% cotton fibres. The afterflame and afterglow times of less than 2 s met the criteria for non-flammability of the knitted fabrics according to the limited flame spread test method. The results show that the investigated blends have the potential to be used for affordable flame-retardant and thermally comfortable knitted fabrics for underwear.

## 1. Introduction

Firefighters, racecar drivers, oil refinery workers, and other professionals who are exposed to the risk of fire are recommended to wear clothing made of flame-resistant textile materials [1]. They often perform intensive physical activity at high environmental thermal radiation, which can reach a very high heat flux of up to 150 kW/m^2^, as is the case in a wildfire, with temperatures of up to 1700 °C [2] (p. 7). Permanent flame resistance is provided only by inherently flame-resistant fibres, but these are usually very expensive. The price of flame-resistant outerwear shell fabrics can be reduced by blending inherently flame-resistant fibres with cheaper fibres, mostly cotton. This usually compromises flame resistance, so such fabrics require additional chemical treatment; e.g., a shell fabric made of an inherently flame-resistant meta-aramid fibre Nomex and cotton must be additionally treated with flame-resistant chemicals if more than 20% cotton is blended [3].

Underwear refers to t-shirts, underpants, bras, and socks, as well as special garments, such as neck cuffs, hoods, and balaclavas, that are used for personal protection [4]. The main function of underwear is to maintain the thermophysiological comfort of the human body. When the human body dissipates excess heat by sweating, the underwear in contact with the skin must manage the sweat in a way that ensures a dry skin surface and allows heat exchange through latent heat transfer. Moisture management must take place as quickly as possible [5] and in close proximity to the skin in order to achieve effective cooling [6] through sweat evaporation. The evaporated sweat must be removed from the firefighter’s clothing; otherwise, it will saturate and condense inside the clothing after some time. If there is higher water vapour pressure in the environment than in the garment, the water vapour will penetrate the garment, where it will accelerate condensation. The heat released during the condensation of water vapour is large and very unfavourable because it increases the risk of skin burns [7].

Regardless of the recommendations, firefighters in particular do not wear flame-resistant underwear but ordinary underwear, mainly made of cheap cotton or polyester [8]. Polyester underwear is particularly popular among firefighters because it is soft, easy to wash, and durable, and because there are several attractive designs available on the market [9,10]. In moderate-temperature environments, good moisture management allows underwear made of blends of cotton or lyocell with elastane to provide good adhesion to the skin and rapid absorption of sweat from the skin [8]. Polyester microfibre underwear is also very effective in removing sweat from the skin by wicking [4]. Cotton and lyocell fibres burn quickly with cool ash residue when in contact with a flame, while polyester fibres burn and melt with accumulating heat. It has been proven that underwear created from blends of polyester (or polyamide or polypropylene) fibres with cotton is not suitable in an extreme temperature environment because melting of the synthetic fibres produces molten plastic material that can easily penetrate the burned structure of the cotton fibres (forming what is called a “char scaffold”), where it continues to burn [9]. This effect was described by Harrocks [10] in 1986. Therefore, the use of underwear made of cotton, thermoplastic fibres, or their blends requires the wearing of much better flame-resistant outerwear to adequately protect the human body [11].

Personal protective flame-resistant underwear available on the market is usually made of flame-resistant meta-aramid fibres, polyamide-imide fibres, wool, modacrylic fibres, or blends of various inherently flame-resistant fibres with hydrophilic cotton or lyocell fibres. Avsec [12] states that polybenzimidazole fibres [13] and oxidised polyacrylonitrile fibres are also suitable for firefighter underwear, but they are not yet found in regular online offerings in the market. Ryklin et al. [14,15] state that polyoxazole fibre is comparable to meta-aramid fibre, which is the most recognisable among inherently flame-resistant fibres. Glombikova and Komarkova [16] studied non-flammable underwear available on the market, including a knitted fabric of 50% polyoxazole/50% viscose FR fibres. Table 1 lists the most popular examples of textile raw materials on the market in online offers, which are used in underwear for different target groups.

Inherently flame-resistant fibres used for flame-resistant underwear (see Table 2) are self-extinguishing, with a minimum LOI (limiting oxygen index) of 26%. They are thermostable and can last up to 190 °C for a long period without deterioration in mechanical properties. Some of them are even stable up to 250 °C and have a higher resistance to ignition in clothing. Their physical properties are comparable to those of cotton and wool: a tenacity of 1.3–5 cN/dtex, a density of 1.27–1.5 g/cm^3^, and a moisture regain of 0.4–15%.

The need in the market for affordable flame-resistant underwear is ubiquitous. Conventional modacrylic/cotton blends are among the cheaper ones. Modacrylic fibres [17] are about 3–4 times cheaper than meta-aramid fibres [18], which are the best-known fibres for flame-resistant underwear on the market. This article focuses on a new fibre blend of a cheap modacrylic fibre, a lyocell fibre that is much more sustainable than cotton [19], and a new polyacrylate fibre with an extreme LOI value of 40% and excellent water absorption. The blends were studied with respect to (1) the different linear density of the modacrylic fibres, (2) the structure of the ring-spun yarns produced by different processes, and (3) the different structure of the knitted fabrics. The quality of the yarns and the wetting time, absorption time, afterflame time, and afterglow time of the fabrics were investigated as important indicators of the suitability of the blends for flame-resistant underwear.
materials-16-04386-t001_Table 1Table 1Examples of fibres available on the market used in underwear.Target GroupUseFibres for Underwearconsumersusuallycotton/elastane, 100% cotton [20]less oftenpolyester, polyamide 6.6, polypropylene [21]wool, bamboo viscose, cellulose modal [22]firefighters,soldiers,police officersusuallycotton/elastane, polyester [12]less often100% meta-aramid [23],blends of meta-aramid with viscose FR or cotton [24],blends of modacrylic with cotton [25,26] or lyocell [27,28] or lyocell/polyamide [29], 100% polyamide-imide,blends of polyamide-imide with viscose FR, wool/viscose FR [30,31],polyoxazole/CO/PES [32], wool/elastane/antistatic fibre [33], chloro fibre/viscose FR/cotton/polyamide [24]racecar drivers and their crewsusually100% meta-aramid [23,34]less often100% wool, 100% polyamide-imide, wool/modacrylic [35]
materials-16-04386-t002_Table 2Table 2Properties of inherently flame-resistant staple fibres suitable for personal protective underwear.FibreTrademarkLOI ^1^(%)T_decomp in air_ ^2^(°C)Tenacity (cN/dtex)Density(g/cm^3^)Moisture Regain(%)Referencemeta-aramidNomex284272.3–4.41.37–1.384–8.3[36]modacrylicProtex 26–33247–2771.6–2.21.35–1.370.4–4.0[37]polyoxazoleArselon 30465>2.81.4310–12[14,38]polyamide-imideKermel30200/3804–4.41.344[33]polybenzimidazolePBI414502.41.4315.0[1]OPAN ^3^Panox>45310 1.61.3910.0[39]^1^ Limiting oxygen index; ^2^ temperature of onset of intensive decomposition (rapid weight loss) in air, defined by thermogravimetric analysis; ^3^ oxidised polyacrylonitrile.

A modacrylic fibre (MAC) is a self-extinguishing synthetic fibre created by copolymerizing acrylonitrile and vinyl chloride that does not melt when heated but carbonizes and forms a char barrier. On the market, there are the fibre Kanekaron, with an LOI of 26–32%, and a special modacrylic fibre, Protex, with an LOI of 33%, developed for personal protective clothing [37]. Knitted fabrics made of modacrylic fibres are flame-resistant and highly abrasion-resistant but prone to pilling and wrinkling. The fabrics can be easily dyed and washed.

A polyacrylate fibre (PAC) was first produced by Acordis in 1988, but, because of its low tenacity (0.12–0.15 cN/dtex) [1], the fibre was used only for nonwovens for a long time. A 2008 patent [40] disclosed a process for developing nontoxic and nearly non-existent fumes produced when PAC fibres are burned. In 2013 [41], a process for modifying a polyacrylate fibre with various nanoparticles to increase its flame resistance and water-absorbing properties was disclosed. In a 2015 patent [42], a process for reducing the red colour of PAC and increasing water absorption was disclosed. In recent years, three new polyacrylate fibres have been introduced to the market, PyroTex [33], Didon [43,44], and Metis [45]. PAC fibres today can achieve an LOI of 40–43% [46] and high moisture and water absorption [47]. A generic name for a polyacrylate fibre is currently being approved by BISFA and FTC. In 2017, an EU regulation defined it as “a fibre formed of cross-linked macromolecules having more than 35 wt.% acrylate groups (acid, light metal salts or esters) and less than 10 wt.% acrylonitrile groups in the chain and up to 15 wt.% nitrogen in the cross-linking” [48]. A polyacrylate fibre in a fibre blend can be quantified according to the methods described in standard ISO 1833 [49].

A modern PAC fibre can be used pure (100%) or in blends. Mills [50] states that polyacrylate fibres included in a two-component fibre blend generally reduce the char length of a fabric exposed to fire in a vertical flammability test. His patent discloses that staple spun yarns created by the ring, air jet, open end, or DREF processes and fabrics created therefrom exhibit improved flame-retardant/heat-resistant properties by adding 5–95 wt.% PAC fibres with an LOI of 35–45% to other fibres in an intimate fibre blend. He states that the addition of PAC fibres also makes yarns and fabrics cheaper. Scientific studies on textiles created from polyacrylate fibres or blends with these fibres are rare. In 2018, Nayak et al. [51] compared knitted fabrics created from 100% PAC fibres with knitted fabrics created from a blend of 93% Nomex, 5% Kevlar, and 2% antistatic fibres. The knitted fabrics created from pure PAC fibres absorbed and retained more water and dried faster than the knitted fabric created from a blend with Nomex fibres. No study has been found that addressed the mechanical properties of fabrics created from pure PAC fibres or from blends with PAC fibres, although low tenacity is a critical property of all PAC fibres. It is about 1–2 cN/dtex [45] and is comparable to merino wool at 0.88–1.5 cN/dtex [52], while it differs from the much stronger cotton fibre at about 2–5 cN/dtex [53].

A lyocell fibre (CLY) is produced from regenerative cellulose in a highly ecological process [54]. CLY can absorb more moisture than CO and allows better moisture management in hot environments [19]. Since CLY is a more sustainable fibre than CO, blends with CLY are also more sustainable than blends with CO. In addition to the standard CLY fibre, a highly porous CLY fibre is on the market that exhibits improved water absorption while minimally compromising its mechanical properties [55]. Underwear is usually made of ring-spun yarns as they have good mechanical properties. Ring-spun yarns can be produced by the conventional, compact, or Sirospun processes [56]. Compared to the conventional ring spinning process, the compact spinning process allows a maximum number of fibres to be spun into the yarn, which improves the properties of the final compact yarns: lower hairiness (free protruding fibres on the surface), greater uniformity, and better mechanical properties compared to conventional ring-spun yarns [57]. Yarns with lower hairiness tend to lint less, and knitted fabrics created from compact yarns have better pilling resistance than those created from conventional ring-spun yarns [58]. The Sirospun process [59] mimics the twisting process in which a single yarn is produced from two rovings. By eliminating the twisting process, the production cost of a yarn can be reduced. The two rovings used in the Sirospun process can be the same or different, i.e., made of different fibres or different colours, which enables special yarn effects [60]. Due to the improved structure, Sirospun yarn is more uniform, has fewer irregularities, is less hairy, and has better mechanical properties than conventional ring-spun yarn. Therefore, fabrics created from Sirospun yarn should have better abrasion resistance, lower pilling tendency, and better washing resistance than fabrics created from conventional ring-spun yarn [61].

Several studies have been conducted to investigate the influence of material composition and fabric structure on the sorption and flammability properties of knitted fabrics. An et al. evaluated the wearing comfort of knitted fabrics made of organic cotton, bamboo-viscose-blended fabric, and soybean-blended fabric in plain, French terry, and 1 × 1 rib structure. They confirmed that material composition and knitted structure are the most important factors for moisture transport and management [62].

Thermal comfort of firefighters is highly dependent on moisture management of the clothing layers closest to the skin. Petrusic et al. studied the influence of fabric structure, composition, and physical properties on moisture transport away from the skin. Tests on liquid moisture transport showed that underwear made of a combination of natural and synthetic fibres transported moisture away from the skin surface faster and more efficiently than cotton underwear, regardless of the type of knit [63].

A wet feel and wet cling can be an important source of sensory discomfort in situations involving heavy perspiration, such as in firefighters’ work environments. To objectively evaluate this aspect of comfort, the Kawabata Evaluation System (KES) was used by Nawaz et al. Seven commercially available knitted fabrics made of different fibre blends (100% wool, 100% cotton, 100% polyester, and various wool blends) in different knitted structures suitable for the skin layer of firefighters’ protective clothing were evaluated first in the untreated and then in the wet condition. The influence of the physical parameters of the fabric, fibre content, fabric construction, and moisture content on the surface properties of the fabric was determined. The study showed that fibre content and fabric structure are the most important parameters affecting fabric surface properties relevant to sensory comfort. Single jersey structures are best suited for next-to-skin use, and 100% wool and wool blended with bamboo provide better sensory comfort compared to 100% cotton or 100% polyester [64].

In their research, Glombikova and Komarkova evaluated the efficiency of non-flammable functional underwear used as a secondary heat barrier in extreme conditions. Five groups of knitted fabrics were analysed for flame resistance and selected physical properties. Double jersey made of 60% FR modacrylic/40% cotton had the least area, circumference, and length of burn marks and had the best overall moisture management capability (OMMC) and cumulative one-way transport capacity (OWTC) values of all fabrics tested [16].

## 2. Materials and Methods

### 2.1. Sample Preparation

All samples used in this research were produced on industrial machines. Two types of modacrylic fibres (MAC 1.0 and MAC 1.7), a lyocell fibre (CLY), and a polyacrylate fibre (PAC) (see Table 3) were used to prepare two intimate fibre blends (see Table 4), from which two rovings with a linear density of 0.33 ktex were spun on a Zinser 670 machine at a take-up speed of 20 m/min (see Table 5).

From the rovings, three types of ring-spun yarns (see Table 6) were spun on a ring spinning machine at a take-up speed of 11 m/min using the conventional, Sirospun, and compact spinning processes. The yarns were used to produce knitted fabrics on a Shima Seiki SES 122 flat knitting machine RT, gauge 12E (see Table 6), which were also tested for their sorption and flame resistance properties. Each knitted fabric was created from two identical yarns that were combined to form a double thread. The same settings were used for all yarns: feed tension, couliering depth (cam settings), and knitted fabric takedown. From each yarn (see Table 5), five knitted fabrics with different structures were produced (see Figure 1).

The knitted structures studied differed in texture, thickness, surface area, extensibility, and compactness/openness, all of which could have an effect on flammability and thermal comfort. The plain knitted structure is a single knitted structure. It is commonly used for underwear, has a smooth surface, and is also the thinnest of all the structures studied. 2:1 rib and 2:1 tuck rib are double knitted structures and therefore thicker than the plain structure. They have repeated and very pronounced vertical ridges. The 2:1 tuck rib is more open than the 2:1 rib due to the combination of tucks and long loops. Both structures are stretchable in width, which allows for a close fit on the body. Single piqué is a single structure and is often used for sportswear because its wavy texture allows for a larger surface area. Tucks make the structure more open and porous. The triple tuck structure has an even more pronounced wavy texture than single piqué and thus a larger surface area. Three consecutive tucks in combination with extended loops make the structure porous and distinctly relief-like. The patterns and control programmes for making the knitted samples were created using the Shima Seiki design system SDS-ONE. The natural colour of the PAC fibres was pink. The rovings, yarns, and knitted fabrics with 15% added PAC fibre were pink as well. In addition, a comparison of knitted fabrics created from blends of 55% modacrylic/15% polyacrylate/30% lyocell with a blend of 65% modacrylic/35% cotton was performed. Modacrylic/cotton is the most affordable fibre blend used for inherently flame-resistant underwear. The flame-resistant blends contain at least 55% modacrylic fibres [65].

### 2.2. Testing of Fibres

Prior to analysis, all fibres were soaked in distilled water for one hour, then vacuum filtered to remove excess water, and air dried. The surface morphology of the fibres was determined from microphotographs taken with a scanning electron microscope JEOL JSM -6060 LV (Jeol, Tokyo, Japan). The IR spectra were recorded with an FT-IR Spectrum GX spectrophotometer (Perkin Elmer, Beaconsfield, UK) using the ATR method in the range 4000–500 cm^−1^ with a resolution of 4 cm^−1^. Each final spectrum represents the average of 16 spectra. The diameter of the fibres was determined using the ImageJ programme (U. S. National Institutes of Health, Bethesda, Maryland, USA) [66]. The linear density of each fibre was determined according to standard ISO 1973:1999 [67] using a Lenzing Vibroscope 400 (Lenzing Instruments, Gampern, Austria). Fibre length was determined according to ISO 6989:1995 [68] by measuring and calculating the average length of 30 individual fibres. Dynamic scanning calorimetry of the fibres was performed using a DSC Q200 instrument (TA Instruments, New Castle, DE, USA) equipped with a liquid nitrogen cooling accessory. Fibres with a mass of 4.4–4.7 mg were heated at a rate of 10 °C/min. Measurements were performed in a nitrogen atmosphere at a flow rate of 50 mL/min using standard aluminium pans with lids. Dynamic thermogravimetry of fibres was performed using a Netzsch STA449 C Jupiter DTA instrument (Netzsch-Gerätebau, Selb, Germany). Fibres of 3 mg were heated at a rate of 10 °C/min in a nitrogen atmosphere. Breaking force and elongation of the fibres were tested on a Vibrodyn 400 (Lenzing Technik, Lenzing, Austria) according to standard ISO 5079:1995 [69] at an initial length of 20 mm (PAC and MAC) and with 10 mm (CLY). The fibres were tested with a preload of 150 mg and a deformation rate of 20 mm/min. The flammability of the fibres was tested using a non-standardised monitoring method that included sensory evaluation of the behaviour of a fibre bundle as it approached the gas burner fire, in the fire, and after removal from the fire, detection of the odour produced during combustion, and the characteristics of the residue.

### 2.3. Testing of Yarns

The surface morphology of the yarns was analysed using a Nikon SMZ 800 stereomicroscope (Nikon Instruments Inc., New York, USA) with a 1× magnification objective and a Nikon D750 camera. In addition, microphotographs were taken with a scanning electron microscope JEOL JSM-6060 LV (Jeol, Tokyo, JP). The yarns were treated with a drop of Cl-Zn-I reagent composed of zinc chloride, ZnCl_2_ (66 g), water (34 g), and potassium iodide, KI (6 g) to stain the cellulose in the lyocell fibres.

The linear density of the yarns was tested according to standard EN ISO 2060:1994 [70] with five skeins for each yarn. The tensile properties of the yarns were determined according to standard ISO 6939:1988 [71] on a Textechno Statimat CU (Textechno, Mönchengladbach, Germany) dynamometer with 100 measurements per yarn. The Uster statistics of the yarns were per-formed on an Uster Tester 4 (Uster Technologies AG, Uster, Switzerland) with five measurements per yarn at a test speed of 200 m/min.

The twist multiplier (*K_t_*) was calculated according to Equation (1):*K_t_* = *T* × √*T_t_*,(1)
where *T* is number of twists per metre (m^−1^) and *T_t_* is the linear density of the yarn (tex).

Moisture regain was determined according to standard ISO/TR 6741:1996 [72] at a temperature of 20 ± 2 °C and a relative humidity of 65 ± 2%. Two measurements were performed for each sample. The water retention value (WRV) was determined according to standard DIN 53814 [73]. A sample of 0.4 g of fibres was soaked in 150 mL of distilled water for one hour, and then the excess water was removed by vacuum filtration and centrifuged at a speed of 3000 rpm for 30 min. The WRV (%) was calculated according to Equation (2):WRV = (*m*_2_ − *m*_1_) × 100/*m*_1_,(2)
where *m*_1_ is the mass of the centrifuged sample and *m*_2_ is the mass of the absolutely dry sample, determined after drying at 105 °C for 4 h.

### 2.4. Testing of Knitted Fabrics

All tests were performed on prewashed samples in an Electrolux EW 1247W household washer–dryer at a temperature of 30 °C with a program for delicate laundry, followed by four rinses, a short spin cycle, and a drying time of 40 min. The samples were then conditioned for 24 h and relaxed in a desiccator under standard conditions.

Wetting was determined by measuring the time required for a piece of fabric to sink completely from the surface layer of water in a beaker. For this purpose, a 3 cm × 3 cm specimen was cut from the fabric sample and placed on the water surface in a 500 mL beaker. The wetting time was estimated as the time interval between the moment of immersion and the moment when the sample sank below the water level [74,75].

Absorption time was determined according to the AATCC standard [76] by measuring the time required to spread one drop of water stained with 0.5 g/L methylene blue (Bezaktiv BLAU S-FR 4/94) (Bezema AG, St Gallen, CH) on the fabric surface. Using a pipette, a drop of water was released onto the fabric sample from a height of 2.5 cm. A stopwatch was used to measure the time from contact of a drop of water with the fabric surface until spillage.

The burning behaviour of the knitted fabrics was tested in a combustion chamber using a limited flame spread method in accordance with standard ISO 15025 [77], taking into account the non-flammability criteria cross-related to standard EN 469 (Protective clothing–Requirements for fire fighter’s protective clothing) and the test method of standard EN 532 [78]. According to standard EN 532, six parallels were drawn for one sample, three in the longitudinal direction (warp/wale) and three in the transverse direction (weft/course). Prior to testing, the samples were pre-washed and conditioned. In addition, DIN 53906 [79] was used for testing the flammability of vertically oriented knitted fabrics.

### 2.5. Hypothesis Testing

The influence of fibre blends and spinning processes on yarn properties was determined by analysis of variance of the samples using the multi-factor ANOVA with the help of the statistical program Statgraphics Centurion XV (UpdateStar GmbH, Berlin, Germany) [80]. A statistical confidence level of 95% was used. When the probability value was *p* ≤ 0.05, the null hypothesis was rejected and statistically significant differences between samples were confirmed; when the value was *p* > 0.05, statistically significant differences between samples were not confirmed. When *p* is equal to 0.05, the *t*-test serves to compare means through pairwise comparisons. The mean values of the samples were also evaluated using the *t*-test, which is similar to normal distribution but accounts for the variability in a small number of measurements to determine whether there are statistically significant differences in the mean value of two samples for the selected characteristic.

## 3. Results and Discussion

### 3.1. Fibres

The IR spectra of the CLY, MAC, and PAC fibres (see Figure 2) show absorption bands in the range 3700–3000 cm^−1^ caused by vibrations of the hydroxyl groups of the water absorbed by the fibres. The IR spectra of the MAC 1.0 and MAC 1.7 fibres are identical, from which it can be concluded that these two fibres were made of the same polymer. The absorption band in the range of 2300–2000 cm^−1^ is due to nitrile groups (-CN) in the MAC fibres. The IR spectrum of the PAC fibre has two characteristic absorption bands, at 1651 cm^−1^, corresponding to the deformation vibrations of hydroxyl groups attached to carbon (C-OH), and at 1559 cm^−1^, corresponding to the symmetric stretching of carboxylic anions (-COO^−^), which are salts of carboxylic acids and indicate a larger number of polyacrylate groups contained by the polyacrylate polymer [47,81].

The two fibre blends (MAC 1.7/PAC/CLY and MAC1.0/PAC/CLY) were created from short staple fibres with an average length of 35–54 mm (see Table 7 and Figure 3). The blends differed in the type of MAC fibres: the MAC 1.0 fibre was a microfibre, and the MAC 1.7 fibre was a coarse fibre, while their tenacity was comparable. The only parameter by which the two blends and the yarns created from them differed was the different linear density of the modacrylic fibre, i.e., the component with the highest weight percentage (55 wt.%) in the blends.

All the fibres except the PAC had good strength above 3 cN/dtex. The PAC fibre also stands out for its extreme length and diameter.

### 3.2. Yarns

In Figure 4, longitudinal views of the yarns are presented. The average diameter of the yarns assessed from SEM images of the longitudinal appearance of the yarn was 185–200 μm. All yarns had some protruding fibres on the surface (see Figure 4 (above)). Lyocell fibres were evenly distributed on the surface of all yarns (see black fibres in Figure 4 (middle)).

The PAC fibres with 15 wt.% in the blends were rare but regularly distributed on the surface along the yarns (see yellow fibres in Figure 4 (below)).

Based on Figure 4, it can be assumed that the MAC, PAC and CLY fibres were homogeneously distributed throughout the yarn. The polyacrylate and lyocell fibres in the yarns could effectively absorb the sweat when in contact with the skin surface and remove it from the skin by diffusion. The modacrylic fibres that encounter sweat can wick it away from the skin through a capillary mechanism.

From the Uster statistics (see Table 8), it appears that the compact yarns were generally the most uneven yarns. The spinning process mainly affected the unevenness, thick places, and neps of the yarns with micromodacrylic fibres; however, of all the yarns, the conventional yarn with MAC 1.7 fibres was the most even.

Conventionally spun yarns (Conv/MAC 1.7 and Conv/MAC 1.0) exhibited higher hairiness, i.e., longer fibres protruding from the yarn surface than compact and Sirospun yarns (see Table 8). The protruding hydrophilic fibres, such as lyocell, can absorb sweat and improve the moisture management of the fabric [82]. On the other hand, these fibres can deteriorate the appearance of a fabric and increase its tendency to pilling.

The yarns with MAC 1.0 were more uneven than the yarns with MAC 1.7 fibres: they had higher unevenness of mass, number of thick places, and neps:the number of thin places was the highest in the compact yarns, making these two yarns more prone to tearing than the other yarns.the number of neps was significantly higher in yarns with MAC 1.0 fibres than in yarns with MAC 1.7. It was statistically demonstrated that the linear density of MAC fibres has an influence on the unevenness of the yarns, i.e., on the number of neps in the yarn. Neps occurred most frequently during carding, namely between the carding drum and the caps, when the caps are too close to the carding drum. The neps in the yarn mainly affect the aesthetic appearance of the knitted fabric.

The yarns were very fine, with linear density ranging from 14.4 to 14.9 tex (see Table 9). Yarns created from the blend with MAC 1.0 fibres had a 1.5% higher linear density than yarns created from the blend with MAC 1.7 fibres. Coarser yarns of about 20 tex are commonly used for underwear [83]. Finer yarns could be used to make thinner underwear with lower thermal resistance, which we consider beneficial for the comfort of underwear in general.

All the yarns had suitable useful mechanical properties, but there were some deviations between them (see Table 9). The yarn tenacity depends on the fibre tenacity (i.e., the substantial tenacity of the fibres) and the number of fibres in a yarn cross-section; it increases with twist up to certain values [84]. Yarns with lower linear density usually have lower tenacity because the number of fibres in a yarn cross-section is lower. The conventional ring-spun yarns had the highest tenacity, while the yarns produced by the Sirospun process had the lowest. Although the MAC 1.0 microfibre had slightly lower tenacity (3.13 cN/dtex) than the standard MAC 1.7 yarn (3.17 cN/dtex), the conventional yarn with MAC 1.0 achieved 13.7% higher and the compact yarn 13.5% higher tenacity than the yarns with MAC 1.7. This can be attributed to the larger number of MAC microfibres in the yarn cross-section and to the larger specific surface area of MAC 1.0 fibres (assessed to 3110 cm^2^/g), which was about 90% higher than that of MAC 1.7 fibres. Microfibres with large specific surface area can form more contacts between fibres and generate higher friction [85]. The Sirospun yarn with the MAC 1.0 fibres had lower tenacity than the Sirospun yarn with the MAC 1.7 fibres, indicating that the Sirospun spinning process did not take advantage of their substantial strength through friction because the Sirospun yarns were probably not sufficiently twisted.

Yarns with MAC 1.0 fibres had on average 19.2% lower initial modulus than yarns with standard MAC 1.7 fibres and were softer than yarns with MAC 1.7 fibres. Conventional ring-spun yarns with MAC 1.0 had the lowest initial modulus.

It was demonstrated at a 95% confidence level that the linear density of modacrylic fibres affected the linear density (*p* = 0.0218), breaking force (*p* = 0.0003), and tenacity (*p* = 0.0003) of the yarns. The spinning processes influenced the breaking force, tenacity (*p* = 0.0011), and elongation at break (*p* = 0.000) of the yarns, but not the linear density (*p* = 0.079).

The linear density of the yarns was slightly more influenced by the linear density of the MAC fibres (F = 17.7) than by the selection of spinning process (F = 9.03), while the breaking force of the yarns was more influenced by the selection of spinning process (F = 102.16) than by the linear density of the modacrylic fibres (F = 23.8). The elongation at break was influenced by the selection of spinning process (F = 37.30) and not by the linear density of the modacrylic fibres (F = 1.69).

The differences in spinning processes require optimization of the machine parameters for each spinning process separately in order to optimize the yarn properties; this is particularly necessary for Sirospun yarns in order to increase the number of twists.

### 3.3. Moisture Regain and Water Sorption

Underwear created from fibres with high moisture regain is usually more comfortable than underwear created from fibres with zero or low moisture regain. Moisture regain depends on the hydrophilic nature of a fibre, i.e., the content of free hydrophilic groups (mainly –OH) in the noncrystalline phase of the fibre. The moisture regain of a fibre blend is the sum of the moisture regain of the individual fibres, taking into account their weight share. Moisture regain is changed by temperature and air humidity; therefore, it is usually compared under standard conditions.

Polyacrylate fibres on the market differ in their molecular structure with respect to the proportion of acrylate and acrylonitrile units [47], which affects moisture regain. The moisture regain of commercially available PAC fibres ranges from 13% [44] to 15% [45]. The PAC fibre used in our study achieved 15.55% (see Figure 5a). This value was higher than for lyocell fibres (11.5%). The moisture regain of the yarns was in the range of 6.0–6.56%, which is close to the moisture regain of cotton (6–8.5%) [52], which is most commonly used for underwear [86].

Water retention value (WRV) expresses the amount of water absorbed in a fibre or textile material after wetting and can be an indication of a fibre’s ability to absorb sweat from the skin. The amount of water retained in the fibre depends on its hydrophilicity and internal voids. Kofler et al. [87] demonstrated that underwear created from a blend of CLY/wool protected firefighters from burns better than underwear created from 100% aramid fibres because the blend of CLY/wool blend absorbed more sweat. The amount of water retained by aramid fibres used for flame-resistant underwear is very low, up to 12% [88].

The WRV of the fibres used ranged from 5.0 to 71.1%. The WRV of the PAC fibre (58.8%) was lower than that of CLY (71.1%) (see Figure 5b).

As with moisture regain, the WRV of a blend depends on the proportion of individual fibres in a blend and the available pores between fibres. The WRY of the yarns ranged from 28.3 to 31.0% (see Figure 5b). The WRV of the yarns was near the WRV of wool (32%) [89]. A weak correlation (r_xy_ = 0.8579) was demonstrated in WRV between samples (roving, yarns) with MAC 1.0 and MAC 1.7, but no correlation was observed in moisture regain.

### 3.4. Wetting Time and Absorption Time

The wetting time is the time period in which the top and bottom surfaces of a fabric have started to become wet [90]. It can help to understand how quickly sweat transferred away from the skin surface. Therefore, a lower wetting time is better. A wetting time of ≤5 s for dry fabric meets the requirements for active sports [74].

The wetting time of the knitted fabrics MAC/CO was 480–960 s, while the wetting time of the yarns from the blends MAC/PAC/CLY had wetting times of 3.4–21.4 s (see Figure 6a). The type of fibre was the most influential parameter on wetting time (*p* = 0.0018): the fabrics created from the blends MAC/PAC/CLY had much lower wetting time than fabrics created from the blend MAC/CO. The poor wettability of the MAC/CO fabrics is most likely due to the lower wettability of cotton compared to CLY and PAC fibres.

When comparing blends of MAC/PAC/CLY, it was statistically demonstrated (*p* = 0.0365) that the linear density of MAC fibres affects wetting time: fabrics with MAC 1.0 wetted more slowly than fabrics with MAC 1.7. It was also statistically demonstrated that the type of spinning process did not affect the wetting time (*p* = 0.11307), but the type of the knitted structure affected the wetting time (*p* = 0.0010): the knitted fabrics with triple tuck (V) and plain (LD) structures had the lowest wetting time, while the fabrics with ribbed structures (RL and R 2:1) had the highest wetting time. No correlation was observed between the wetting time and the mass per unit area of fabrics (r_xy_ = 0.2074).

Wetting time depends on the wettability of a fabric surface (i.e., the ability of a liquid to spread and adhere to the surface of a textile material, surface texture) and on the diffusion of the liquid from the surface through the fabric, with the thickness and structure of a knitted fabric having an influence. A rough, relief-like knitted surface provides more contact points for liquids, potentially improving wettability. The plain and the single piqué structures are single knitted structures created on a single-bed knitting machine, while the 2:1 rib, 2:1 tuck rib, and triple tuck are double knitted structures created on a double-bed knitting machine and therefore likely have greater thickness and thus greater potential for diffusion of liquid from the surface through the fabric. It should also be noted that the combination of influencing factors, such as material composition, yarn and fabric structure, and processing, can have complex and interactive effects on the wettability of knitted fabrics. In the case of the knitted structures studied, the rough surface of the triple tuck structure with many contact points and the fabric thickness due to the relief-like structure contributed to the higher wettability. In contrast, a smoother surface of the plain knitted structure has a smaller contact area, yet the liquid spreads easily on the fabric due to the low contact angle, indicating good wettability.

The absorption time of a water drop through a knitted fabric is the time it takes for the water drop to spread on the surface of the knitted fabric. The absorption time for the knitted fabrics from MAC/CO was 480–840 s and was much higher than that for the knitted fabrics from the blends MAC/PAC/CLY, which was 3.4–14.6 s (see Figure 6b). Fabrics created from the blend MAC/PAC/CLY were much better at keeping the skin dry during intense physical activity or in hot environments than knitted fabrics created from a blend of MAC/CO. Knitted fabrics created from the blends with MAC 1.0 fibre had a higher wetting time (5.6–14.6 s) than knitted fabrics created from the blends with standard MAC 1.7 fibre (3.4–7.4 s). The absorption time was the highest for the plain structure (LD) and shortest for the triple tuck structure (V).

It was statistically demonstrated that the absorption time was influenced by the linear density of MAC fibres (*p* = 0.00002), by the spinning process (*p* = 0.00022), and by the structure of the knitted fabrics (*p* = 0.0057).

The triple tuck structure (V) exhibited the lowest wetting and absorption time and is expected to have the best moisture management, while the plain structure (LV), which is normally used for underwear, was found to have poorer moisture management.

Wettability refers to how well a liquid spreads and adheres to the surface of a textile material. Absorption, on the other hand, refers to the ability of a textile material to absorb a liquid and hold it in its structure. It is influenced by factors such as porosity, pore size, density, and surface area of the textile, as well as capillary action in the fibres or pores. The relationship between wettability and absorption is that wettability can affect the absorption behaviour of a textile material. A highly wettable surface can lead to good liquid distribution, which can allow better penetration into the textile structure, resulting in increased absorption. On the other hand, a textile material may have high wettability, meaning that a liquid can easily spread and wet its surface, but may not necessarily have high absorption if the fibre composition and/or fabric structure does not favour retention of liquid. For example, dense and tightly knitted fabrics have limited liquid absorption due to their smaller pore size.

The triple tuck structure consists of three consecutive tucks combined with elongated loops in the wale direction, which is reflected in a greater number and size of pores compared to the other structures studied. In addition, the triple tuck structure has a corrugated surface that is larger than the surface of a plain structure, resulting in good wettability and absorption capacity. All the studied tuck structures, the 2:1 tuck rib, the single piqué, and the triple tuck, are more porous than the plain and the 2:1 rib structure, which consist only of short loops. The wettability of the plain structure studied was high due to the smooth surface, while the low porosity of the structure consisting only of loops (no tucks) resulted in high absorption times.

### 3.5. Thermal Properties and Burning Behaviour

The DSC spectra (where the samples were heated to temperatures up to 300 °C) of the CLY and PAC fibres (see Figure 7a) only show endothermic peaks around 100 °C, which can be attributed to the release of capillary-bound water due to the hydrophilic behaviour of the cellulose fibres. At about 250 °C, the DCS curves begin to rise, indicating the exothermic peaks that would appear above 300 °C for both CLY and PAC fibres, as stated by Carillo [91] and Molnar [65]. For the MAC 1.7 and the MAC 1.0 fibres, the spectra are almost identical. They show small endothermic peaks at about 210 °C, related to melting since the modacrylic fibre is a low crystalline material [92], and an intense exothermic peak at 250 °C, related to chemical changes including crosslinking, decomposition, and oxidation of the modacrylic polymer, accompanied by a high mass loss, also observed in the thermogravimetric analysis (TGA) spectra (Figure 7b).

According to TGA analysis, PAC fibres were stable up to 200 °C, MAC fibres up to 220 °C, and CLY fibres up to 250 °C, with minimal weight loss for all fibres. Temperatures above 200 °C were critical for the MAC fibres, while rapid mass loss was observed for the CLY fibre above 300 °C. The PAC fibres lost their significant mass above 350 °C. In the 200–300 °C range, the MAC fibre lost almost 30% of its mass, while the CLY and PAC fibres lost only 15%.

The measurements of the thermal properties were completed in a nitrogen atmosphere. In an oxygen atmosphere or in air, the loss of mass above 300 °C is higher, as is the case with the meta-aramid fibre Nomex [36].

The results of the simple visual burning test of the MAC, PAC, and CLY fibres (see Table 10) confirmed the self-extinguishing nature of the MAC and PAC fibres [93]. The flame-resistant properties of MAC fibres are due to the polymer composition, in which an added comonomer (e.g., vinyl chloride) provides the fibres low flammability characteristics. No melting or dripping of any of the fibres near the flame and no sticky residue after burning were observed. The shrinking of the MAC fibre near the flame could favour its applications above all in blends [88]. CLY fibres burned completely in a flame, and, after removing the flame, only ashes remained.

During the design of the experiment and prior to the design of the structure of the knitted samples, it was assumed that a very pronounced vertical rib structure could strongly influence the burning behaviour in the vertical direction compared to the burning behaviour in the transverse direction since the rib structure should allow the flame spread in the wale direction, i.e., in the direction of the ribs, while, in the course directions, i.e., perpendicular to the ribs, the flame spread would be hindered. It was also assumed that the wave-like texture of the triple tuck structure may hinder the flame spread in both directions. The plain knitted structure could be the least flame-resistant among the studied structures due to its smooth surface. The vertical direction of the loop limbs could have an effect on the stronger flame spread in the wale direction compared to the course direction, where the loop limbs could form a barrier.

Standard ISO 15025 [77] is intended for testing protective clothing according to a limited flame spread, where non-flammability criteria are recorded: up to a maximum of 2 s of burning after the burner is removed (i.e., afterflame time) and up to a maximum of 2 s of glow after the flame is extinguished (i.e., afterglow time). The afterflame time and the afterglow time of all the knitted fabrics created in triple tuck (V) and 2:1 tuck rib (RL) structures (see Figure 8) were less than 2 s for all knitted fabrics from the two blends MAC/PAC/CLY and also from the MAC/CO blends, without any holes, melting, or dripping. These results confirmed that the knitted fabrics in triple tuck and 2:1 tuck rib have the potential for manufacturing firefighters’ personal protective clothing and that the addition of 15% PAC fibres to the blends undoubtedly had some influence.

Knitted fabrics with incorporated tucks that have a pronounced wave texture, such as the triple tuck structure, or longitudinal ridges, such as the 2:1 tuck rib structure, perfectly meet the non-flammability criteria and at the same time are comfortable due to their wettability and absorbency (triple tuck structure) and their stretchability due to the folded ribs (2:1 tuck rib structure).

The method for testing flammability according to standard DIN 53906 is generally not intended for firefighters’ garments but for testing other textiles where the tested specimens were suspended vertically. The test method was chosen because it provided interesting results and to compare it with the test method of ISO 15025, which has already been discussed.

The main difference between DIN 53906 and ISO 15025 is the position in which the gas burner is placed on the tested sample with a flame. In the case of the DIN 53906 standard, the burner is placed vertically under the tested sample directly at the bottom edge, while the ISO 15025 standard specifies the gas burner to be placed perpendicular to the surface of the tested sample, and, as mentioned above, this is specifically intended for testing materials for firefighters’ protective clothing. In both cases, the tested specimens are vertically oriented and exposed to the flame for 10 s.

In accordance with ISO 15025, the requirements for flame-resistant textiles refer to the average afterflame time, which should be ≤2 s, and the average smouldering time (afterglow time) after removal of the burner flame ≤2 s. None of the specimens should burn to the top of the specimen, no hole should be created through any layer of the specimen except in the top layer of the multilayer material, and the molten residue should not burn in any of the specimens. During the vertical flammability test (see Figure 9), none of the knitted fabrics showed a hole and no combustible embers were produced. The knitted fabrics from the blend with MAC 1.0 fibre showed better flame resistance than the knitted fabrics from the blends with MAC 1.7 fibre. The 2:1 rib structure (R 2:1) from Conv/MAC 1.7, the Siro/MAC 1.7 blend, and the Comp/MAC 1.0 yarns had the best flame resistance, and the plain structure (LD) had the worst. It was confirmed that the vertical ribs and the distinct texture prevent burning and smouldering in the transverse direction due to the pronounced relief knitted structure, which forms a barrier. The smooth surface of the plain structure does not form a barrier and therefore offers little protection against burning.

To test the influence of the type of MAC fibre, spinning process, and knitted fabric structure on the flammability of knitted fabrics, the sum of the afterflame and afterglow times in the course and wale directions of each knitted fabric was used in the multi-factor ANOVA. It was found that only the structure of the knitted fabrics had a statistically significant effect on the afterflame time and afterglow time at the 95.0% confidence level as the *p*-value was 0.0262, i.e., less than 0.05.

## 4. Conclusions

Affordable personal protective flame-resistant underwear available on the market is usually created from blends of modacrylic fibres with cotton or lyocell. In our research, we have confirmed that knitted fabrics composed of modacrylic/cotton blends meet the flame resistance requirements for personal protective clothing, but we have also found that the 35% cotton content in the blend is insufficient to ensure rapid wetting and water absorption, which can lead to poor sweat removal from the skin.

Therefore, we proposed a blend of 55% modacrylic fibres, 30% sustainable lyocell fibres, and 15% polyacrylate fibres, which has an extreme LOI value of 40% and an excellent water retention value close to that of lyocell fibres. Based on the TGA measurements performed, all fibre blend components are stable up to 200 °C, modacrylic even up to 220 °C and lyocell up to 250 °C. Moreover, modern synthetic clothing made of microfibres is known to be breathable (water-vapour-permeable) and soft to the touch. Therefore, one of the blends studied contained modacrylic macrofibres and the other standard modacrylic fibres to determine if there were differences in sorption properties and flame resistance between them. The yarns were produced using different ring spinning processes, i.e., conventional, compact, and Sirospun, determine which yarn has the best performance properties. Knitted fabrics in five different structures were created from all the yarns, and the influence of their structure on flammability and sorption properties was also studied.

Spinning processes affected the hairiness, Uster values, unevenness, and tensile properties of the yarns. The conventional ring-spun yarn with MAC 1.0 fibres had the best mechanical properties, the best evenness, and the best softness (the lowest modulus of elasticity). 

We found that the knitted fabrics containing modacrylic microfibres were worse in terms of wetting rate and also in terms of water absorption rate. It was also found that the choice of spinning process does not affect the wetting and absorption time of the knitted fabrics. We also found that the knitted fabrics containing modacrylic microfibres were as flame-resistant as the knitted fabrics containing standard modacrylic fibres.

The type of knitted structure affected the wetting time, absorption time, and flammability. The knitted fabrics with triple tuck (V) and plain (LD) structures had the lowest wetting time, while the knitted fabrics with ribbed structure had the highest wetting time. The absorption time was the highest for the plain structure (LD) and shortest for the triple tuck structure (V).

The flammability of the knitted structures was not affected by the linear density of the MAC fibres, the spinning process, or the type of knitted structure, and all met the criteria of non-flammability according to standard ISO 15025. The afterflame time and afterglow time are within the permissible limits of the standard requirements (less than 2 s), and no burning or glowing holes are formed in the structures, and no dripping or moulding occurs. We can confirm that the small amount of polyacrylate fibres randomly distributed in the yarn cross-section in the blend with modacrylic and lyocell fibres significantly improves the flame resistance of the fibres, yarns, and knitted fabrics, as well as the rapid sweat absorption—two of the most important properties for knitted underwear for firefighters.

The best flame-resistant knitted structure was the 2:1 rib (R 2:1) created from Conv/MAC 1.7, Siro/MAC 1.7, and Comp/MAC 1.0 yarns.

On the other hand, the triple tuck structure with pronounced wave-like texture from Comp/MAC 1.7 is both flame-resistant and stretchable, as well as highly wettable and absorbent, which are also desirable properties for underwear.

Finally, it should be noted that the comparison of the MAC/PAC/CLY blends with the conventional underwear fibre blend MAC/CO has shown that the wetting time and absorption time for the MAC/PAC/CLY knitted fabrics were much lower, so they have the potential to better keep the skin dry during intense physical activity or in hot environments.

## Figures and Tables

**Figure 1 materials-16-04386-f001:**
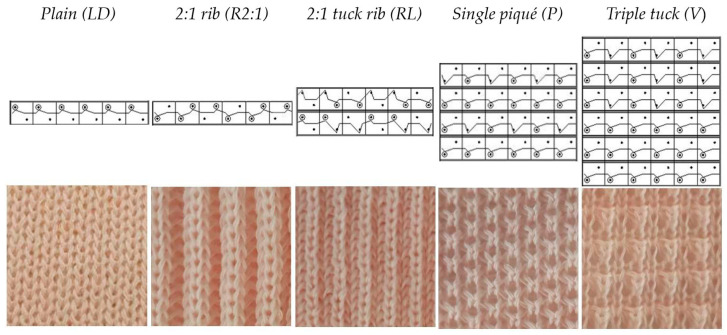
Knitted fabrics: structures’ notations and appearance. (The abbreviations of the knitting structures are given in the brackets).

**Figure 2 materials-16-04386-f002:**
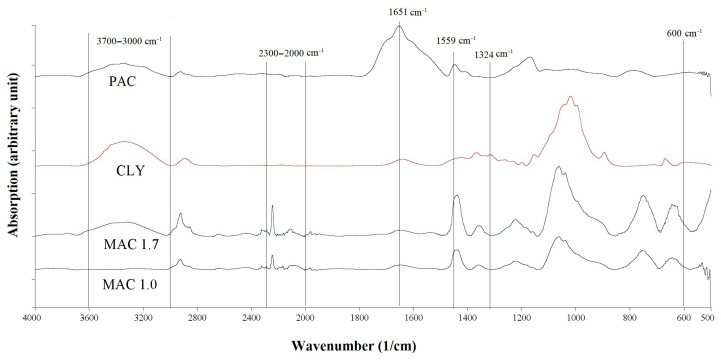
FT-IR spectra of MAC, CLY, and PAC fibres.

**Figure 3 materials-16-04386-f003:**
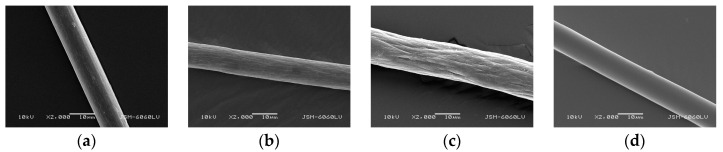
Microphotographs of the used fibres at a magnification of 2000×. (**a**) Modacrylic 1.7 dtex (MAC 1.7). (**b**) Modacrylic 1.0 dtex (MAC 1.0). (**c**) Polyacrylate (PAC). (**d**) Lyocell (CLY).

**Figure 4 materials-16-04386-f004:**
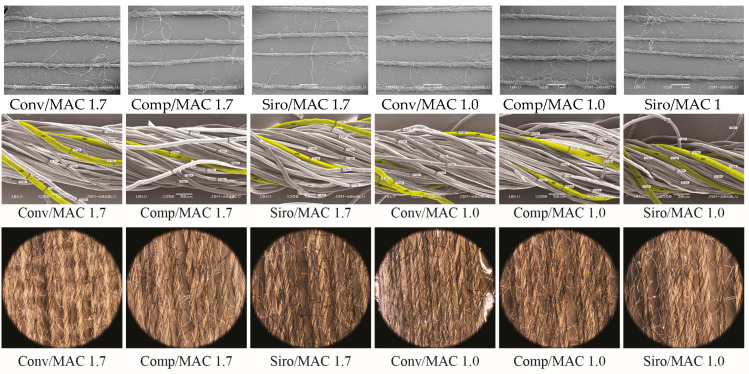
Morphology of spun yarns: (**above**) the scanning electron microscope view of yarns, magnification 20×; (**middle**) the stereo microscope view of yarns with black-coloured lyocell fibres; (**bellow**) the scanning electron microscope view of yarns with yellow-coloured PAC fibres, magnification 350×.

**Figure 5 materials-16-04386-f005:**
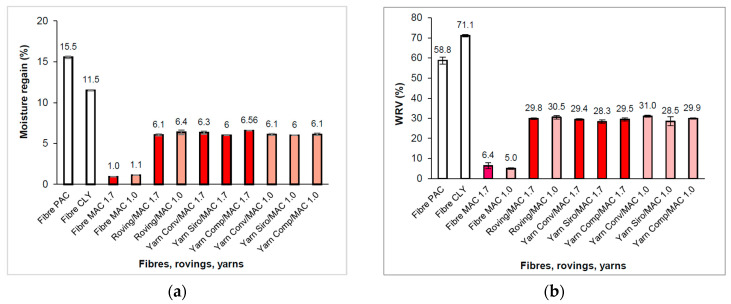
Fibres, rovings, and yarns: (**a**) moisture regain; (**b**) water retention value (WRV).

**Figure 6 materials-16-04386-f006:**
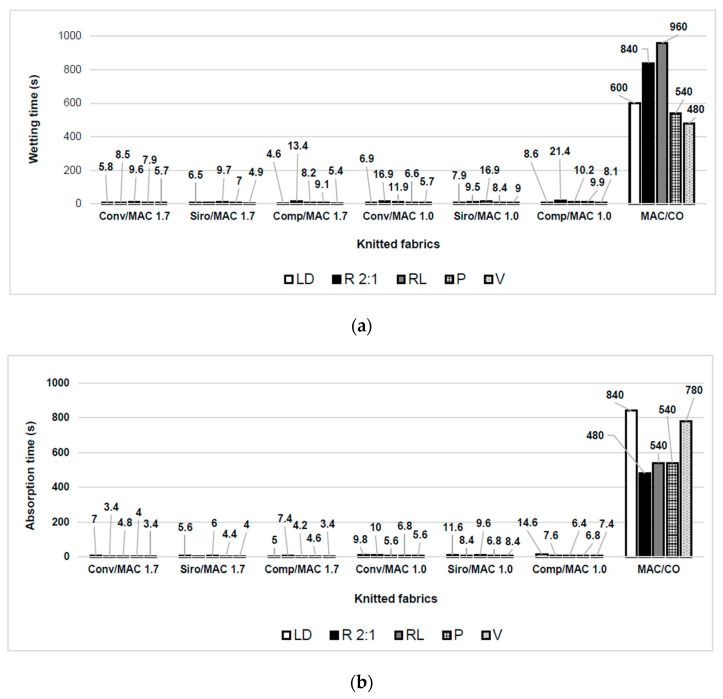
Fabrics with different knitted structures: (**a**) wetting time; (**b**) absorption time.

**Figure 7 materials-16-04386-f007:**
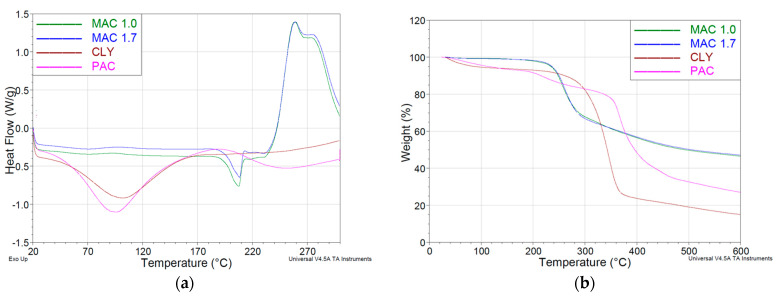
Fibres: (**a**) differential scanning calorimetry; (**b**) thermogravimetric analysis.

**Figure 8 materials-16-04386-f008:**
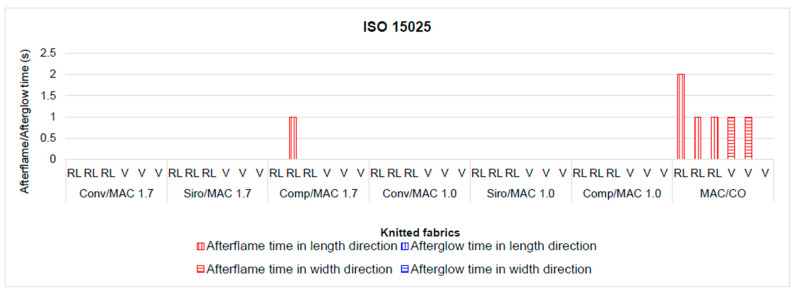
Fabrics with different knitted structures. Afterflame and afterglow times according to ISO 15025.

**Figure 9 materials-16-04386-f009:**
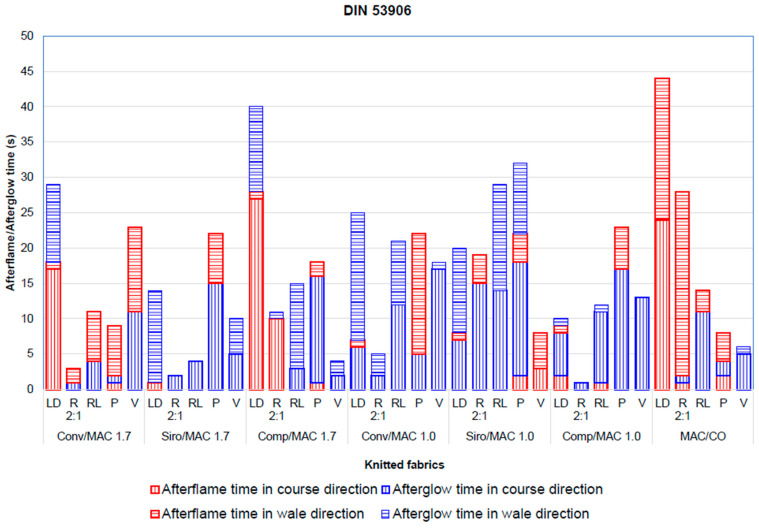
Fabrics with different knitted structures. Afterflame and afterglow time according to DIN 53906.

**Table 3 materials-16-04386-t003:** Fibres’ characteristics specified by suppliers.

Fibre	Linear Density (dtex)	Tenacity (cN/tex)	LOI (%)	Moisture Regain (%)	Supplier
MAC 1.7	1.7	29.2	32.5	3–3.5	Formosa Plastics Corp.
MAC 1.0	1.0	29.2	32.5	3–3.5	Formosa Plastics Corp.
PAC	2.5	14–20	42	11	Technical Fibres Tecstar S.L.
CLY	1.25	>34	<20	8–14	Lenzing AG

**Table 4 materials-16-04386-t004:** Fibre blends.

Fibre Blend	Fibre Component 1		Fibre Component 2		Fibre Component 3	
	Type	Content (%)	Type	Content (%)	Type	Content (%)
MAC 1.7/PAC/CLY	MAC 1.7	55	PAC	15	CLY	30
MAC 1.0/PAC/CLY	MAC 1.0	55	PAC	15	CLY	30

**Table 5 materials-16-04386-t005:** Yarns.

Yarn	Roving	Spinning Process	Spinning Machine	Ring Diameter (mm)	Linear Density (tex)	Twist (m^−1^)
Conv/MAC 1.7	MAC 1.7/PAC/CLY	Conventional	Zinser RM350	40	14.7	1000
Comp/MAC 1.7	MAC 1.7/PAC/CLY	Compact	Zinser RM351	40	14.7	1000
Siro/MAC 1.7	MAC 1.7/PAC/CLY	Sirospun	Zinser RM350	45	14.7	1100
Conv/MAC 1.0	MAC 1.0/PAC/CLY	Conventional	Zinser RM350	40	14.7	1000
Comp/MAC 1.0	MAC 1.0/PAC/CLY	Compact	Zinser RM351	40	14.7	1000
Siro/MAC 1.0	MAC 1.0/PAC/CLY	Sirospun	Zinser RM350	45	14.7	1100

**Table 6 materials-16-04386-t006:** Knitted samples’ descriptions according to yarn type and fibre blend.

Knitted Fabric ^1^	Yarn	Fibre Blend
Knit/Conv/MAC 1.7	Conv/MAC 1.7	MAC 1.7/PAC/CLY
Knit/Comp/MAC 1.7	Comp/MAC 1.7	MAC 1.7/PAC/CLY
Knit/Siro/MAC 1.7	Siro/MAC 1.7	MAC 1.7/PAC/CLY
Knit/Conv/MAC 1.0	Conv/MAC 1.0	MAC 1.0/PAC/CLY
Knit/Comp/MAC 1.0	Comp/MAC 1.0	MAC 1.0/PAC/CLY
Knit/Siro/MAC 1.0	Siro/MAC 1.0	MAC 1.0/PAC/CLY
Knit/MAC 1.5/CO	Conventional ring spinning	65% MAC 1.5 dtex/35% cotton LS

^1^ From each yarn, knitted fabrics in five different structures were created (see Figure 1).

**Table 7 materials-16-04386-t007:** Linear density and mechanical properties of fibres.

Fibre	Length (mm)	Diameter (μm)	Linear Density (dtex)	Breaking Force (cN)	Tenacity (cN/dtex)	Breaking Elongation (%)	Young Modulus (MPa)
MAC 1.7	36.5	9.0	1.71 ± 0.04	5.42 ± 0.22	3.17	35.92 ± 0.44	217.5 ± 8.59
MAC 1.0	35.4	10.4	1.05 ± 0.02	3.29 ± 0.12	3.13	36.10 ± 0.70	104.5 ± 8.92
PAC	49.7	10.1	2.54 ± 0.06	2.89 ± 0.05	1.14	27.68 ± 1.75	76.2 ± 2.52
CLY	35.9	13.6	1.45 ± 0.03	5.51 ± 0.23	3.80	11.69 ± 0.30	488 ± 26.09

**Table 8 materials-16-04386-t008:** Uster statistics for yarns.

Yarn	Um ^1^ (%)	CVm ^2^ (%)	Thin Places ^3^ (km^−1^)	Thick Places ^4^ (km^−1^)	Neps ^5^ (km^−1^)	Hairiness
Conv/MAC 1.7	11.8 ± 0.08	15.00 ± 0.13	12 ± 4.77	106 ± 9.14	77 ± 10.44	4.43 ± 0.06
Comp/MAC 1.7	12.80 ± 0.47	16.28 ± 0.61	62 ± 29.90	229 ± 79.43	136 ± 33.20	4.14 ± 0.05
Siro/MAC 1.7	12.15 ± 0.09	15.39 ± 0.12	31 ± 6.60	131 ± 9.54	82 ± 9.30	4.26 ± 0.03
Conv/MAC 1.0	12.43 ± 0.03	15.85 ± 0.13	18 ± 5.38	229 ± 6.96	265 ± 27.62	4.37 ± 0.05
Comp/MAC 1.0	13.14 ± 0.21	17.01 ± 0.32	43 ± 14.88	425 ± 39.94	466 ± 61.76	4.36 ± 0.10
Siro/MAC 1.0	12.17 ± 0.07	15.53 ± 0.10	10 ± 5.70	187 ± 14.88	171 ± 10.89	4.36 ± 0.05

^1^ Unevenness; ^2^ coefficients of variation of mass; ^3^ number of yarn thin places −50%; ^4^ number of yarn thick places +50%; ^5^ number of yarn neps +200%.

**Table 9 materials-16-04386-t009:** Linear density and mechanical properties of yarns.

Yarn	Linear Density (tex)	Twist Multiplier	Breaking Force (cN)	Tenacity (cN/tex)	Breaking Elongation (%)	Young Modulus (MPa)
Conv/MAC 1.7	14.64 ± 0.07	3826.2	133.49 ± 1.68	9.12	6.38 ± 0.12	2.50
Siro/MAC 1.7	14.58 ± 0.04	3818.4	126.88 ± 1.78	8.70	6.265 ± 0.13	2.25
Comp/MAC 1.7	14.44 ± 0.11	3800.0	143.07 ± 2.14	9.90	6.87 ± 0.10	2.38
Conv/MAC 1.0	14.87 ± 0.03	3856.2	154.26 ± 2.33	10.37	7.71 ± 0.13	1.92
Siro/MAC 1.0	14.60 ± 0.08	3821.0	113.24 ± 2.02	7.76	5.54 ± 0.13	2.34
Comp/MAC 1.0	14.70 ± 0.06	3834.1	165.27 ± 2.79	11.24	6.70 ± 0.145	2.20

**Table 10 materials-16-04386-t010:** Flammability characteristics of fibres.

Fibre	Near a Flame	In a Flame	After Removal from a Flame	Residue	Smell
PAC	No changes	Burn	Extinguished	Slightly charred the remnant, which quickly disintegrated	Unpleasant smell of plastic
CLY	No changes	Burn	Burn completely	Ash	Smell of burnt paper
MAC	Shrinkage	Burn	Extinguished	A hard black charred residue that does not disintegrate	Unpleasant smell of plastic

## Data Availability

The data presented in this study are available on request from the corresponding author.

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
