# Peer review of "Quality of Fine Yarns from Modacrylic/Polyacrylate/Lyocell Blends Intended for Affordable Flame-Resistant Underwear"

_materials, 2023, doi:10.3390/ma16124386_

Round 1
Reviewer 1 Report
The article presents an original study on the influence on the wetting management and the flame reaction of different knitted fabrics based on different yarns with variations on the spinning processes and the type of modacrylic fibers mixed with other polyacrylate and lyocell fibers. The state of the art is sufficiently detailed with numerous references. The description of materials and characterization methods is complete even if some information could be added (see comments on the attached file). The results are generally clearly analyzed. However, the authors often remain on observations without giving some reasons (at least hypothesis) of the differences between the various cases presented (for example, there is a lack of explanation for the difference on the reaction to the flame due to the knitted structures, due to the types of yarn or to the fiber modacrylic).

Author Response
Response to Reviewer 1 Comments
Dear reviewer, we are very grateful for all your comments that allow us to improve article!
Point 1: The description of materials and characterization methods is complete even if some information could be added (see comments on the attached file).
Response 1: The knitted structures studied differed in texture, thickness, surface area, extensibility, compactness/openness, all of which could have an effect on flammability and thermal comfort. The plain knitted structure is a single knitted structure. It is commonly used for underwear, has a smooth surface, and is also the thinnest of all the structures studied. 2:1 rib and 2:1 tuck rib are double knitted structures and therefore thicker than the plain structure. They have repeated and very pronounced vertical ridges. The 2:1 tuck rib is more open than the 2:1 rib due to the combination of tucks and long loops. Both structures are stretchable in width, which allows for a close fit on the body. Single piqué is a single structure and is often used for sportswear because its wavy texture allows for a larger surface area. Tucks make the structure more open and porous. The triple tuck structure has an even more pronounced wavy texture than single piqué and thus a larger surface area. Three consecutive tucks in combination with extended loops make the structure porous and distinctly relief-like.
Point 2: The results are generally clearly analyzed. However, the authors often remain on observations without giving some reasons (at least hypothesis) of the differences between the various cases presented (for example, there is a lack of explanation for the difference on the reaction to the flame due to the knitted structures, due to the types of yarn or to the fiber modacrylic).
Response 2: During the design of the experiment and prior to the design of the structure of the knitted samples, it was assumed that a very pronounced vertical rib structure could strongly influence the burning behaviour in the vertical direction compared to the burning behaviour in the transverse direction, since the rib structure should allow the flame spread in the wale direction, i.e., in the direction of the ribs, while in the course directions, i.e., perpendicular to the ribs, the flame spread would be hindered. It was also assumed that the wave-like texture of the triple tuck structure may hinder the flame spread in both directions. The plain knitted structure could be the least flame resistant among the studied structures due to its smooth surface. The vertical direction of the loop limbs could have an effect on the stronger flame spread in the wale direction compared to the course direction where the loop limbs could form a barrier.
Knitted fabrics with incorporated tucks that have a pronounced wave texture, such as the triple tuck structure, or longitudinal ridges such as the 2:1 tuck rib structure, perfectly meet the non-flammability criteria and at the same time are comfortable due to their wettability and absorbency (triple tuck structure) and their stretchability due to the folded ribs (2:1 tuck rib structure).
For testing the influence of type of MAC fibre, spinning process, and structure of knitted fabric on the flammability of the knitted fabrics, the sum of the afterflame and afterglow times in course and wale directions of each knitted fabric was used for multi-fcator ANOVA. It was found out that only the structure of the knitted fabrics had a statistically significant effect on the afterflame and afterglow time at the 95.0% confidence level, as the p-value was 0.0262, i.e. lower than 0.05.
Affordable personal protective flame resistant underwear available on the market is usually made of blends of modacrylic fibres with cotton or lyocell. In our research, we found that knitted fabrics made of modacrylic/cotton meet the non-flammability requirements for personal protective clothing. However, we also found that the 35% cotton content in the blend was not sufficient to ensure rapid wetting and rapid water absorption and could result in poor sweat removal from the skin.
Therefore, we proposed a blend of 55% modacrylic fibres, 30% sustainable lyocell fibres, and 15% polyacrylate fibres, which has an extreme LOI value of 40% and excellent water retention value which is close to that of lyocell fibres. Based on the TGA measurements carried out, all fibre blend components are stable up to 200 °C, modacrylic even up to 220 °C and lyocell up to 250 °C. In addition, modern synthetic clothing made of microfibers is known to be breathable (water vapour permeable) and soft to the touch. Therefore, one of the blends we studied contained modacrylic macrofibres and the other standard modacrylic fibres to see if there were differences in sorption properties and flame resistance between them. The yarns were produced with different ring spinning processes, i.e. conventional, compact and Sirospun, to find out which yarn has the best utility properties. Knitted fabrics in five different structures were made from all the yarns, and the influence of structure on flammability and sorption properties was also studied.
Point 3: (pdf article) Authors should comment the choice of these different structures.
Response 3: We partially answered the point 3 in response 2. Several studies have been conducted to investigate the influence of material composition and fabric structure on the sorption and flammability properties of knitted fabric (mentioned references 62,79).
Knitted fabrics in five different weft structures have been selected according to the existing use for underwear and according to the differences in the topography of the structures..
Point 4: (pdf article) Give the type of atmosphere for thermal analyses.
Response 4: Dynamic thermogravimetry of fibres was performed using a Netzsch STA449 C Jupiter DTA instrument (Netzsch-Gerätebau, DE). Fibres of 3 mg were heated at a rate of 10 °C/min in a nitrogen atmosphere.
Point 5: (pdf article) Cl-Zn-J
Response 5: The yarns were treated with a drop of Cl-Zn-I reagent composed of zinc chloride, ZnCl2 (66 g), water (34 g), and potassium iodide, KI (6 g) to stain the cellulose in the lyocell fibres.
Point 6: Correlation between wetting time and thickness of the knitted fabrics ?
Response 6: Regarding the effect of fabric thickness, it should be noted that fabric thickness is measured under load (standard ISO 5084:1996). When the load is applied to the fabric, the fibers and yarns within the fabric structure are compressed. This compression causes the fabric to become denser, reducing its thickness. The extent of this compression and thickness reduction depends on several factors, including material composition, fabric structure, and inherent properties of the fabric. Since the examined knitted structures consist of different types of knitted structure elements (loops, tucks, miss yarns) in different combinations, the thickness reduction due to standard loading may also vary. In addition, the plain and single piqué structures are single knitted structures produced on a single-bed knitting machine, while 2:1 rib, 2:1 tuck rib, and triple tuck are double knitted structures produced on a double-bed knitting machine and are therefore likely to have a higher thickness. As a result, it is difficult to establish a realistic correlation between the thickness of the fabric in the unloaded state and the wettability by measurements based on standard test methods.
Point 7: Authors should comment a little more in detail the influence of the presence of tucks in knitted structures on the wetting properties.
Response 7: Wetting time depends on the wettability of a fabric surface (i.e. the ability of a liquid to spread and adhere to the surface of a textile material, surface texture) and on the diffusion of the liquid from the surface through the fabric, with the thickness and structure of a knitted fabric having an influence. A rough, relief-like knitted surface provides more contact points for liquids, potentially improving wettability. The plain and the single piqué structures are single knitted structures made on a single-bed knitting machine, while the 2:1 rib, 2:1 tuck rib, and triple tuck are double knitted structures made on a double-bed knitting machine and therefore likely have greater thickness and thus greater potential for diffusion of liquid from the surface through the fabric. It should also be noted that the combination of influencing factors such as material composition, yarn and fabric structure, and processing can have complex and interactive effects on the wettability of knitted fabrics. In the case of the knitted structures studied, the rough surface of the triple tuck structure with many contact points and the fabric thickness due to the relief-like structure contributed to the higher wettability. In contrast, a smoother surface of the plain knitted structure has a smaller contact area, yet the liquid spreads easily on the fabric due to the low contact angle, indicating good wettability.
Point 8: Authors should recall the difference between the two standards which could explain the difference between the results.
Response 8: The main difference between DIN 53906 and ISO 15025 is in the position of placing the gas burner with a flame on the tested sample. In the case of DIN 53906 standard, the burner is placed vertically below the tested sample, directly to the bottom edge, while ISO 15025 standard specifies the gas burner to be placed perpendicular to the surface of the tested sample and is, as mentioned before, specifically intended for testing the materials intended for fire fighter's protective clothing. In both cases, the tested specimens are vertically oriented and exposed to the flame for 10 s.

Reviewer 2 Report
Peer Review for the Article "Quality of Fine Yarns from Modacrylic/Polyacrylate/Lyocell Blends Intended for Affordable Flame Resistant Underwear"
The article produced by Rijavec et al. explores the influence of the linear density of modacrylic fibers on the quality of fine yarns and knitted fabrics. The study investigates three different spinning processes (conventional, Sirospun, and compact ring spinning) and five stitching configurations (lain, 2:1 rib, 2:1 tuck rib, single pique, and triple tuck) to develop affordable flame-resistant underwear.
Overall, it is necessary to standardize the units throughout the text, either in full (seconds) or abbreviated (s). Care should be taken to ensure consistency in this regard.
Since the need for low-flammability underwear may not be obvious, it is recommended to emphasize the importance of this topic in the abstract of the article. Additionally, for didactic purposes, a clear order of information should be presented in the abstract, such as a justification statement, objective statement, methodology overview (2-3 sentences), and a summary of results and conclusions (2-3 sentences).
There is insufficient evidence for the authors to claim that "The new fiber blends have proven to be suitable for affordable underwear." The study focused on the raw materials of the underwear, and no product validation was conducted. It would be more appropriate to state that there are indications that affordable and high-quality underwear could be produced using the studied fiber blends.
Regarding the introduction, the amount of text explaining the contextual background appears to be appropriate. It should be noted that the author provides a significant amount of clear results (quantitative or qualitative) to establish the required quality standards for the raw materials used in underwear production. The absence of a reference material for comparison strengthens the need for the study's findings.
It is important to mention how the results shown in Table 3 were obtained. If the results were provided by the manufacturers of the materials, this should be clarified.
Regarding Table 7, it should be verified if its content can be presented in the form of a figure according to the journal's guidelines.
The term "statistical reliability" should be replaced with "confidence level," as the p-value is not a measure of probability. When p is equal to 0.05, the t-test serves to compare means through pairwise comparisons. The explanation provided by the author contains terminological inconsistencies, and it is recommended to review other studies that have performed this type of analysis.
In Figure 1, "arbitrary units" should be singular since it is expected that all spectra are reported in the same unit. "Wavelength number" should be replaced with "wavenumber."
Why were statistical tests not used to compare the results presented in Tables 8, 9, 10, and others? It is important to include this analysis. The same applies to Figure 4 and all other possible pairwise mean comparisons.
It is suggested to exclude the graph titles in Figures 5, 6, and 8. The axis titles provide sufficient information about the data. Furthermore, it is recommended to combine these figures to reduce the unnecessarily large number of figures.
The usefulness of Figure 9 is unclear. No comments were provided in the text, and the samples appear to be identical.
There is a typo in the vertical axis title of Figure 10. Please review all figures and tables for such issues.
There is very little or almost no explanation of the obtained results, particularly in terms of discussing the results. This aspect requires significant improvement, utilizing relevant literature sources.
The conclusions are extensive and unclear. It is important to summarize the main findings in a concise manner, extracting key statements that effectively conclude the study. As they currently stand, the conclusions appear as a collection of randomly presented sentences.
Overall, the article has potential, but it requires substantial revisions and improvements in terms of standardization, clarity, result interpretation, and conclusion formulation. Taking into account the suggestions provided, the article can significantly enhance its contribution to the field of flame-resistant underwear materials.
Peer Review for the Article "Quality of Fine Yarns from Modacrylic/Polyacrylate/Lyocell Blends Intended for Affordable Flame Resistant Underwear"
The article produced by Rijavec et al. explores the influence of the linear density of modacrylic fibers on the quality of fine yarns and knitted fabrics. The study investigates three different spinning processes (conventional, Sirospun, and compact ring spinning) and five stitching configurations (lain, 2:1 rib, 2:1 tuck rib, single pique, and triple tuck) to develop affordable flame-resistant underwear.
Overall, it is necessary to standardize the units throughout the text, either in full (seconds) or abbreviated (s). Care should be taken to ensure consistency in this regard.
Since the need for low-flammability underwear may not be obvious, it is recommended to emphasize the importance of this topic in the abstract of the article. Additionally, for didactic purposes, a clear order of information should be presented in the abstract, such as a justification statement, objective statement, methodology overview (2-3 sentences), and a summary of results and conclusions (2-3 sentences).
There is insufficient evidence for the authors to claim that "The new fiber blends have proven to be suitable for affordable underwear." The study focused on the raw materials of the underwear, and no product validation was conducted. It would be more appropriate to state that there are indications that affordable and high-quality underwear could be produced using the studied fiber blends.
Regarding the introduction, the amount of text explaining the contextual background appears to be appropriate. It should be noted that the author provides a significant amount of clear results (quantitative or qualitative) to establish the required quality standards for the raw materials used in underwear production. The absence of a reference material for comparison strengthens the need for the study's findings.
It is important to mention how the results shown in Table 3 were obtained. If the results were provided by the manufacturers of the materials, this should be clarified.
Regarding Table 7, it should be verified if its content can be presented in the form of a figure according to the journal's guidelines.
The term "statistical reliability" should be replaced with "confidence level," as the p-value is not a measure of probability. When p is equal to 0.05, the t-test serves to compare means through pairwise comparisons. The explanation provided by the author contains terminological inconsistencies, and it is recommended to review other studies that have performed this type of analysis.
In Figure 1, "arbitrary units" should be singular since it is expected that all spectra are reported in the same unit. "Wavelength number" should be replaced with "wavenumber."
Why were statistical tests not used to compare the results presented in Tables 8, 9, 10, and others? It is important to include this analysis. The same applies to Figure 4 and all other possible pairwise mean comparisons.
It is suggested to exclude the graph titles in Figures 5, 6, and 8. The axis titles provide sufficient information about the data. Furthermore, it is recommended to combine these figures to reduce the unnecessarily large number of figures.
The usefulness of Figure 9 is unclear. No comments were provided in the text, and the samples appear to be identical.
There is a typo in the vertical axis title of Figure 10. Please review all figures and tables for such issues.
There is very little or almost no explanation of the obtained results, particularly in terms of discussing the results. This aspect requires significant improvement, utilizing relevant literature sources.
The conclusions are extensive and unclear. It is important to summarize the main findings in a concise manner, extracting key statements that effectively conclude the study. As they currently stand, the conclusions appear as a collection of randomly presented sentences.
Overall, the article has potential, but it requires substantial revisions and improvements in terms of standardization, clarity, result interpretation, and conclusion formulation. Taking into account the suggestions provided, the article can significantly enhance its contribution to the field of flame-resistant underwear materials.
Author Response
Response to Reviewer 2 Comments
Dear reviewer, we are very grateful for all your comments that allow us to improve article!
Point 1: Overall, it is necessary to standardize the units throughout the text, either in full (seconds) abbreviated (s). Care should be taken to ensure consistency in this regard.
Response 1: Corrected. For seconds is now used strictly only “s”.
Point 2: Since the need for low-flammability underwear may not be obvious, it is recommended to emphasize the importance of this topic in the abstract of the article.
Response 2: Corrected.
The examination of the flammability of underwear is a subject that is often overlooked and is rarely on the list of textile material for fire safety testing. However, especially in the case of professionals that are exposed to the risk of fire, it is important to study flammability of outerwear garments as their direct contact with the skin can have a decisive importance in the extent and degree of skin burns.
Point 3: Additionally, for didactic purposes, a clear order of information should be presented in the abstract, such as a justification statement, objective statement, methodology overview (2-3 sentences), and a summary of results and conclusions (2-3 sentences).
Response 3: Corrected as follows:
- justification statement: The examination of the flammability of underwear is a subject that is often overlooked and is rarely on the list of textile material for fire safety testing. However, especially in the case of professionals that are exposed to the risk of fire, it is important to study flammability of outerwear garments as their direct contact with the skin can have a decisive importance in the extent and degree of skin burns.
- objective statement: This research is focused on the suitability of affordable blends of 55 wt.% modacrylic, 15 wt.% polyacrylate, and 30 wt.% lyocell fibers which have the potential to be used for flame resistant underwear. The impact of the modacrylic fibre linear density, the conventional, Sirospun, and compact ring spinning processes, as well as the structure of knitted fabrics (plain, 2:1 rib, 2:1 tuck rib, single pique, and triple tuck) on their properties required for thermal comfort in high-temperature environments were investigated.
- methodology overview (2-3 sentences): To assess the desired suitability the scanning electron and optical microscopy, FT-IR spectroscopy, mechanical testing, moisture regain, water sorption, wettability, absorption, DSC, TGA and the flammability were tested.
- a summary of results and conclusions (2-3 sentences) The wetting time (5−14.6 s) and water absorption time (4.6−21.4 s) of the knitted fabrics have shown excellent ability to transport and absorb water compared to the knitted fabrics made from conventional blend of 65% modacrylic and 35% cotton fibres. The after-burn and after-glow time of less than 2 s meet the non-flammability criteria of the limited flame spread test method of the knitted fabrics. The results indicate that high-quality flame resistant knitted fabrics intended for underwear could be produced using the studied fiber blends.
Point 4: There is insufficient evidence for the authors to claim that "The new fiber blends have proven to be suitable for affordable underwear." The study focused on the raw materials of the underwear, and no product validation was conducted. It would be more appropriate to state that there are indications that affordable and high-quality underwear could be produced using the studied fiber blends.
Response 4: Corrected. The afterflame time and the afterglow time of all the knitted fabrics made in triple tuck (V) and 2:1 tuck rib (RL) structures (see Figure 8) were less than 2 s for all knitted fabrics from the two blends MAC/PAC/CLY and also from the blends MAC/CO, without any holes, molting or dripping. These results confirmed that the knitted fabrics in triple tuck and 2:1 tuck rib are suitable for manufacturing firefighters' personal protective underwear as they undoubtedlyshowed fast wetting and absorption of water, that is important for thermal comfort.
Point 5: Regarding the introduction, the amount of text explaining the contextual background appears to be appropriate. It should be noted that the author provides a significant amount of clear results (quantitative or qualitative) to establish the required quality standards for the raw materials used in underwear production. The absence of are reference material for comparison strengthens the need for the study's findings.
Response 5: We do not find the reviewer's comment, "The absence of are reference material for comparison strengthens the need for the study's findings." entirely clear. The design experiment included analysis of the new flame resistant yarns and a commercially available reference yarn from MAC/CO used for protective underwear available on the market (see Table 1). In addition, a comprehensive overview of the state of research is given in Chapter 1.
Point 6: It is important to mention how the results shown in Table 3 were obtained. If the results were provided by the manufacturers of the materials, this should be clarified.
Response 6: Corrected. Fibres properties given in the Table 3 have been provided by suppliers that are now corrected in the title of the column six (last column).
Point 7: Regarding Table 7, it should be verified if its content can be presented in the form of a figure according to the journal's guidelines.
Response 7: Agree. Done, as suggested.
Point 8: The term "statistical reliability" should be replaced with "confidence level," as the p-value is not a measure of probability. When p is equal to 0.05,the t-test serves to compare means through pairwise comparisons. The explanation provided by the author contains terminological inconsistencies, and it is recommended to review other studies that have performed this type of analysis.
Response 8: Corrected as follows:
A statistical confidence level of 95% was used. When the probability value was less or equal 0.05 (p £ 0.05), the null hypothesis was rejected and statistically significant differences between samples were confirmed; when the value was greater than 0.005 (p > 0.05), statistically significant differences between samples were not confirmed. When p is equal to 0.05, the t-test serves to compare means through pairwise comparisons. The mean values of the samples were evaluated using the t-test, which is similar to normal distribution but accounts for the variability of a small number of measurements to determine whether there are statistically significant differences in the mean value of two samples for the selected characteristic.
Point 9: In Figure 1, "arbitrary units" should be singular since it is expected that all spectra are reported in the same unit. "Wavelength number" should be replaced with "wavenumber."
Response 9: Corrected. Now, this is Figure 2.
Point 10: Why were statistical tests not used to compare the results presented in Tables 8, 9, 10, and others? Itis important to include this analysis. The same applies to Figure 4 and all other possible pair wisemean comparisons.
Response 10: The influence of the linear density of modacrylic fibres (standard, micro), the influence of the type of yarn (conventional, compact, Sirospun), and the influence of the type of knitting fabrics (plain, 2:1 rib, single pique, triple tuck) on the properties have been analysed as follow:
- According to the multi-factor ANOVA all results given in the Tables 9 (now Table 8) and 10 (now Table 9) were evaluated from the point of view of two factors: linear density of modacrylic fibres and spinning process:
- As Uster analyses were made in a laboratory of a spinning mill Predilnica Litija in Slovenia, according to their regular procedure used in the lab every day, the individual measurements were not disposed, but already statistically processed values (mean values, standard deviations, coefficient of variation). We used t-test to check the influence of the two factors (linear density, spinning process) on the Uster statistics:
Corrected in the article:
The spinning process influenced on the Uster statistics of the yarns with micro modacrylic fibres unevenness (Um), thick places and neps, but the conventional yarn with MAC 1.7 fibres had the lowest unevenness among all the yarns.
- Differences in moisture regain (MR) and water retention value (WRV) (Chapter 3.3) were small between yarns. TMR & WRV mainly depend on the proportion of individual types of fibres in the blends and fibres MR & WRV.
Correlation analyses showed:
- no correlation in MR between samples (roving, yarns) with MAC 1.0 and MAC 1.7 have been evidenced (rxy = 0.269).
- a weak correlation in WRV between samples (roving, yarns) with MAC 1.0 and MAC 1.7 have been evidenced (rxy = 0.8579).
Corrected in the article:
A weak correlation (rxy = 0.8579) in WRV between samples (roving, yarns) with MAC 1.0 and MAC 1.7 have been evidenced, but in moisture regain there were evidenced no correlation.
- Wetting Time and Absorption Time
By comparing blends of MAC/PAC/CLY, it was statistically proved (p = 0.0365) that the linear density of MAC fibres affects the wetting time: fabrics with MAC 1.0 wetted more slowly than fabrics with MAC 1.7.
It was also statistically proved that the type of the spinning process did not affect the wetting time (p = 0.11307), but the type of the knitted structure affected the wetting time (p = 0.0010): the knitted fabrics with triple tuck (V) and plain (LD) structures had the lowest wetting time, while the fabrics with ribbed structures (RL and R 2:1) had the highest wetting time. No correlation was proved between the wetting time and the mass per unit area of fabrics (rxy = 0.2074).
- Afterflame and afterglow time according to DIN 53906
Influence of three factors (the type of MAC fibre, the type of spinning process and the type of knitted fabric) on the afterflame time and afterglow time in course and wale directions were tested by using multi-factor ANOVA analyses. The results show the value p < 0.05 only for type of knitted fabrics (p = 0.0262), that had a statistically significant effect on afterflame and afterglow time at the 95,0% confidence level:
Analysis of Variance for Afterflame Time and Afterglow Time - Type III Sums of Squares
Corrected in the article:
To test the influence of the type of MAC fibre, spinning process and knitted fabric structure on the flammability of knitted fabrics, the sum of the afterflame and afterglow times in the course and wale directions of each knitted fabric was used in the multi-factor ANOVA. It was found that only the structure of the knitted fabrics had a statistically significant effect on the afterflame time and afterglow time at the 95.0% confidence level, as the p-value was 0.0262, i.e., less than 0.05.
Point 11: It is suggested to exclude the graph titles in Figures 5, 6, and 8. The axis titles provide sufficient information about the data. Furthermore, it is recommended to combine these figures to reduce the unnecessarily large number of figures.
Response 11: The Figures 5 and 6 are now included in a new Figure 6. The graph titles have been excluded in these figures.
The titles in the Figures 8 and 9 have not been excluded, because there is no duplication with the titles on ordinates. We are sure that it is welcome for the reader if, with the title, he is quickly reminded of the method by which the material was analysed.
Point 12: The usefulness of Figure 9 is unclear. No comments were provided in the text, and the samples appear to be identical.
Response 12: Agree. The Figure 9 has been deleted.
Point 13: There is a typo in the vertical axis title of Figure10. Please review all figures and tables for such issues.
Response 13: Corrected.
Point 14: There is very little or almost no explanation of the obtained results, particularly in terms of discussing the results. This aspect requires significant improvement, utilizing relevant literature sources.
Response 14: In the revised version, numerous corrections, additional explanations, and discussions have been added. The abstract and conclusions have been rewritten. Relevant references are included in the state of research in Introduction, and also in Results and discussion.
Point 15: The conclusions are extensive and unclear. It is important to summarize the main findings in a concise manner, extracting key statements that effectively conclude the study. As they currently stand, the conclusions appear as a collection of randomly presented sentences.
Response 15: Conclusions have been re-written, as follows:
Affordable personal protective flame resistant underwear available on the market is usually made from blends of modacrylic fibres with cotton or lyocell. In our research, we have confirmed that knitted fabrics made of modacrylic/cotton meet the flame resistance requirements for personal protective clothing, but we have also found that the 35% cotton content in the blend is insufficient to ensure rapid wetting and water absorption, which can lead to poor sweat removal from the skin.
Therefore, we proposed a blend of 55% modacrylic fibres, 30% sustainable lyocell fibres and 15% polyacrylate fibres, which has an extreme LOI value of 40% and an excellent water retention value close to that of lyocell fibres. Based on the TGA measurements performed, all fibre blend components are stable up to 200 °C, modacrylic even up to 220 °C and lyocell up to 250 °C. Moreover, modern synthetic clothing made of microfibers is known to be breathable (water vapour permeable) and soft to the touch. Therefore, one of the blends studied contained modacrylic macrofibres and the other standard modacrylic fibres to determine if there were differences in sorption properties and flame resistance between them. The yarns were produced using different ring spinning processes, i.e. conventional, compact and Sirospun, to find out which yarn has the best performance properties. Knitted fabrics in five different structures were made from all the yarns, and the influence of their structure on flammability and sorption properties was also studied.
Spinning processes affected the hairiness, Uster values, unevenness and tensile properties of the yarns. The conventional ring-spun yarn with MAC 1.0 fibres had the best mechanical properties, the best evenness, and the best softness (the lowest modulus of elasticity ).
We have found that the knitted fabrics containing modacrylic microfibres were worse in terms of wetting rate and also in terms of water absorption rate. It was also found that the choice of spinning process does not affect the wetting and absorption time of the knitted fabrics. We also found that the knitted fabrics containing modacrylic microfibres were as flame resistant as the knitted fabrics containing standard modacrylic fibres.
The type of knitted structure affected the wetting time, absorption time, and flammability. The knitted fabrics with triple tuck (V) and plain (LD) structures had the lowest wetting time, while the knitted fabrics with ribbed structure had the highest wetting time. The absorption time was the highest for the plain structure (LD) and shortest for the triple tuck structure (V).
The flammability of the knitted structures was not affected by the linear density of the MAC fibres, the spinning process, or the type of knitted structure, and all met the criteria of non-flammability according to the standard ISO 15025. The afterflame time and afterglow time are within the permissible limits of the standard requirements (less than 2 s), and no burning or glowing holes are formed in the structures, and no dripping or moulding occurs. We can confirm that the small amount of polyacrylate fibres randomly distributed in the yarn cross-section in the blend with modacrylic and lyocell fibres significantly improves the flame resistance of the fibres, yarns and knitted fabrics, as well as the rapid sweat absorption - two of the most important properties for knitted underwear for firefighters.
The best flame resistant knitted structure was the 2:1 rib (R 2:1) made from Conv/MAC 1.7, Siro/MAC 1.7 and Comp/MAC 1.0 yarns.
On the other hand, the triple tuck structure with pronounced wave-like texture from Comp/MAC 1.7 is both flame resistant and stretchable, as well as highly wettable and absorbent, which are also desirable properties for underwear.
Finally, it should be noted that the comparison of the MAC/PAC/CLY blends with the conventional underwear fibre blend MAC/CO has shown that the wetting time and absorption time for the MAC/PAC/CLY knitted fabrics were much lower, so they have the potential to better keep the skin dry during intense physical activity or in hot environments.

Round 2
Reviewer 2 Report
I have reviewed the previous review report, along with the author responses and the revised manuscript, and I am pleased to note that the authors have made significant improvements based on the feedback provided. The revised version appears to be well-refined and ready for publication.
It seems good to me.